# Volatiles from *Pseudomonas palleroniana* Strain B-BH16-1 Suppress Aflatoxin Production and Growth of *Aspergillus flavu*s on *Coix lacryma-jobi* during Storage

**DOI:** 10.3390/toxins15010077

**Published:** 2023-01-14

**Authors:** Shihua Zhou, Qing-Song Yuan, Xiaoai Wang, Weike Jiang, Xiaohong Ou, Changgui Yang, Yanping Gao, Yanhong Wang, Lanping Guo, Luqi Huang, Tao Zhou

**Affiliations:** 1Resource Institute for Chinese & Ethnic Materia Medica, Guizhou University of Traditional Chinese Medicine, Guiyang 550025, China; 2National Resource Center for Chinese Materia Medica, China Academy of Chinese Medical Sciences, State Key Laboratory of Dao-di Herbs, Beijing 100700, China

**Keywords:** semen coicis, *Aspergillus flavus*, aflatoxins, *Pseudomonas palleroniana*, volatiles, endosphere bacteria

## Abstract

Semen coicis is not only a traditional Chinese medicine (TCM), but also a typical food in China, with significant medical and healthcare value. Because semen coicis is rich in starch and oil, it can be easily contaminated with *Aspergillus flavus* and its aflatoxins (AFs). Preventing and controlling the contamination of semen coicis with *Aspergillus flavus* and its aflatoxins is vital to ensuring its safety as a drug and as a food. In this study, the endosphere bacteria *Pseudomonas palleroniana* strain B-BH16-1 produced volatiles that strongly inhibited the mycelial growth and spore formation activity of *A. flavus*. Gas chromatography–mass spectrometry profiling revealed three volatiles emitted from B-BH16-1, of which 1-undecene was the most abundant. We obtained authentic reference standards for these three volatiles; these significantly reduced mycelial growth and sporulation in *Aspergillus*, with dimethyl disulfide showing the most robust inhibitory activity. Strain B-BH16-1 was able to completely inhibit the biosynthesis of aflatoxins in semen coicis samples during storage by emitting volatile bioactive components. The microscope revealed severely damaged mycelia and a complete lack of sporulation. This newly identified plant endophyte bacterium was able to strongly inhibit the sporulation and growth of *Aspergillus* and the synthesis of associated mycotoxins, thus not only providing valuable information regarding an efficient potential strategy for the prevention of *A. flavus* contamination in TCM and food, but potentially also serving as a reference in the control of toxic fungi.

## 1. Introduction

*Coix lacryma-jobi* L. belongs to the Gramineae family. Its semen coicis is not only used in traditional Chinese medicine (TCM) but is also a typical food in China. Semen coicis has significant medical and healthcare value for treating edema, dampness arthralgia, lung carbuncles, intestinal carbuncles, verruca, and cancer [1,2,3]. However, semen coicis can be easily contaminated by *Aspergillus flavus* and its aflatoxins (AFs), because it is rich in starch and oil. Many investigations have shown that the detection rate of AFs in semen coicis and its products ranges from 25% to 100.0% in the Huanghuai River basin, the Yangtze River basin, and the southeast coastal areas. Rates of pollution with AFB_1_ and AFB_2_ have reached serious levels as high as 69.7% and 36.0%, respectively [4]. In addition, the rate at which AFs in semen coicis exceed the maximum permissible levels has reached 62% [4]. Based on the high degree of pollution of semen coicis with *Aspergillus flavus* and its toxins, the 2020 edition of the Chinese Pharmacopoeia stipulated maximum limits of aflatoxin and *Aspergillus flavus*. Preventing and controlling the contamination of *Aspergillus flavus* and its aflatoxins in semen coicis is of great significance to ensuring its safety as a food.

*A. flavus* is one of the most common saprophytic aerobic pathogens in nature, and is able to infect many essential crops (i.e., peanuts, rice, maize, and wheat) [5,6,7,8] and materials used in TCM (e.g., semen coicis, lotus seeds, and jujube) [9,10,11,12,13], both before and after harvest. In addition, it can produce more than 20 highly toxic carcinogenic, teratogenic, and mutagenic aflatoxins, posing a great threat to human health worldwide and leading to severe food and drug safety risks. Among these, AFB_1_, AFB_2_, AFG_1_, AFG_2_, AFM_1_, and AFM_2_ are classified in humans as Group I carcinogens by the World Health Organization (WHO) and the International Agency for Research (IARC) [9,14]. According to previous investigations, about 5 billion people worldwide are exposed to high levels of *A. flavus* and aflatoxin contamination. In addition to causing severe human health issues and food and drug safety risks, the presence of *A. flavus* and its aflatoxins can also lead to economic losses. In the USA, with high-end storage technology and equipment, crop losses can reach up to USD 500 million annually, of which corn loss alone accounts for as much as USD 160 million [15]. However, in developing countries, due to their lack of storage technology and equipment, the economic losses caused by *A. flavus* and its aflatoxins are far more significant than those in developed countries.

The fundamental elimination of infection with *A. flavus* and its toxins is a crucial issue to solve in order to ensure the safety of food and TCM. The prevention and control of *A. flavus* and its aflatoxins during storage mainly depends on the management of storage conditions, the use of chemical agents, and the removal of diseased grain. However, these management methods are highly dependent on the equipment and the workforce, and are highly time and energy consuming, as well as being costly. The application of chemical agents can cause potentially harmful fungicide residues and environmental pollution, and their intensive use can cause a rapid increase in the emergence of resistant *A. flavus* strains [16]. Existing control measures have been unable to fulfill the requirements of ensuring the safety of food and TCM. Therefore, screening new, safe, and effective antifungal resources is a primary task for the management of *A. flavus* and its aflatoxins both before and after harvest.

It is widely known that microbes are rich, easy to culture, and produce various metabiotic substances, which have been found to be effective against many fungal pathogens. Therefore, strategies have been developed and used to control *A. flavus* and its aflatoxins [7,17]. For instance, *Bacillus subtilis* and *B. cereus* have been proven to inhibit the growth of *A. flavus* and the production of aflatoxins [18,19]. Some species of *Candida* and *Pichia* also have inhibitory activities on *A. flavus* and its aflatoxins [20]. In addition, various volatiles from *Hericium erinaceus* [21] and plants such as *Zingiber officinale* [22], citronella [23], thyme [24], and cinnamon [25] have been reported to be helpful in the management of many pathogens both before and after harvest. Previous research conducted in the last decade has shown that the volatiles from bacteria exert inhibitory activity against various plant pathogens [16,26]. For example, *Shewanella algae*, producing dimethyl trisulfide and 2,4-bis (1,1-dimethylethyl)-phenol, can significantly inhibit the growth of *A. flavus* in vitro, along with the germination of conidia, and prevent aflatoxin production in stored peanut and maize [27]. Although the volatile substances that have been reported to prevent and control *A. flavus* inhibit the growth of mycelia and the germination of spores, very few bio-active volatiles have been found to be effective at inhibiting the formation of spores.

In this study, we isolated an endosphere bacteria—*Pseudomonas palleroniana* strain B-BH16-1—which produces volatiles with strong inhibitory activities against *A. flavus* and its aflatoxins. Our objective was to determine the antifungal effect of the *P. palleroniana* strain B-BH16-1 on the mycelial growth and spore formation of *A. flavus* and to evaluate the inhibitory activity of B-BH16-1 on aflatoxin biosynthesis, as well as the structures and antifungal activities of the volatiles emitted by B-BH16-1 against *A. flavus*. The B-BH16-1 strain produced three volatiles—including dimethyl disulfide, 1-undecene, and squalene—that strongly inhibited mycelial growth and spore formation in *A. flavus*. In addition, strain B-BH16-1 was able to completely inhibit aflatoxin biosynthesis by emitting volatile bioactive components. This newly identified plant endophyte bacterium is able to strongly inhibit the formation of *Aspergillus* spores and the growth of mycelia, which has a practical value for application in blocking the spread of *A. flavus,* which contaminates food and materials used for TCM.

## 2. Results

### 2.1. Identification of Antifungal Bacterial Strain B-BH16-1

The sequence of V3V4 rRNA and genome were used for genetic identification. The V3V4 rRNA sequence of B-BH16-1 was aligned with known strain sequences in the GeneBank database. The homologous strains with more than 97% similarity were selected to construct the phylogenetical tree. The results showed that strain B-BH16-1 could be assigned to *Pseudomonas*, and was tremendously homologous with *P*. *palleroniana*, *P*. *paoe*, *P*. *tolaas*, and *P*. *flourescens* species, while the highest homology, that with *P*. *palleroniana*, was 99% (Figure 1). In addition, on the basis of genome similarity analysis, it was also found that the ANI of B-BH16-1 with *P*. *palleroniana* was greater than with other species. The biochemical analysis further proved that strain B-BH16-1 showed the greatest homology with strain *P*. *palleroniana* in the BIOLOG MicroStation™ database (Biolog Inc, Hayward, CA, USA). A total of 26 carbon resources could be used, including α-d-glucose, sucrose, mannitol, d-sorbitol, glycerol so on. The same reaction was also shown when evaluating tolerance to different concentrations of NaCl, pH, and temperature. These results proved that B-BH16-1 might be belong to the *P*. *palleroniana* species.

### 2.2. Antifungal Activity of P. palleroniana Strain B-BH16-1 via the Emission of Volatiles and the Secretion of Diffusible Substances

The dual culture method and face-to-face (FTF) dual culture method were used to detect the antifungal activity of B-BH16-1. On the basis of the dual culture test, we found that the mycelia of *A. flavus* in the control group grew rapidly and luxuriantly on the PDA plate for 5 days (Figure 2A,C). In addition, the semidiameter of the mycelium in the control treatment reached up to 2.2 cm. Meanwhile, in the B-BH16-16-amended group, B-BH16-1 was able to significantly inhibit the growth of the mycelia of *A. flavus*. The semidiameter of the *A. flavus* mycelia was less than 1.3 cm. The inhibition rate reached 40.1% (Figure 2B). In addition, B-BH16-1 formed a prominent bacteriostatic circle (Figure 2A). The FTF dual culture test presented similar inhibition of *A. flavus* in the dual culture test (Figure 2A,D), with the inhibition rate reaching 70.0% (Figure 2B). These results showed that B-BH16-1 emits volatiles and secretes diffusible components in order to control *A. flavus*.

### 2.3. Inhibitory Effect of P. palleroniana Strain B-BH16-1 on the Spore Formation and Mycelium Malformation of A. flavus

The spore is a vital organ for *A. flavus* transmission and diffusion. Therefore, we tested the effect of B-BH16-1 on spore formation using an FTF dual culture test. Microscopic observations found that *P. palleroniana* strain B-BH16-1 significantly inhibited spore peduncle development and sporangium formation (Figure 3A). It was further found by counting the spores that the number of spores in the *P. palleroniana* strain B-BH16-1 group was considerably lower than that in the control group (Figure 3B). The expression of genes was analyzed to better understand the effect of volatiles on spore development. Three genes with essential functions in the spore development were selected for RT-qPCR analysis, and it was found that all genes were downregulated by volatiles from the *P. palleroniana* strain B-BH16-1 compared to the control treatment (Figure 3C,D). In the particular case of the *RodB* gene, expression was downregulated at 10 dpi and 20 dpi by up to 42 and 45 fold, respectively. The expressions of *FlbC* and *FlbD* genes were downregulated at 10 dpi by up to 14 and 10 fold, respectively, and exhibited significant differences compared to the control treatment.

In addition, microscopic observation also showed that the *P. palleroniana* strain B-BH16-1 caused significant mycelium malformation in *A. flavus*; the mycelium was twisted and dented, showing a decrease in the rigidity of the cell membrane (Figure 3A). Then, we detected the expression of three genes essential to cell membrane synthesis, and found that the *P. palleroniana* strain B-BH16-1 significantly inhibited the expression of these genes (Figure 3E–G). The expressions of *Csd1*, *Csd2*, and *Sam1* were greatly downregulated at 10 dpi by up to 3, 5, and 3 fold, respectively.

### 2.4. Biocontrol Activity of the P. palleroniana Strain on A. flavus Growth and Aflatoxin Biosynthesis in Storage

To prove that the *P. palleroniana* strain B-BH16-1 exerts biocontrol activity against *A. flavus* and the biosynthesis of its toxins in storage, methods for detecting biocontrol activity were adopted as described by Gong et al. As shown in Figure 4A, the symptoms of disease in coix seeds at 10 dpi were more severe than at 20 dpi in the control. The kernel surface in the control treatment was abundantly covered in *A. flavus* conidia and mycelia. The disease incidence of coix seeds reached up to 100%, both at 10 dpi and at 20 dpi. However, no disease symptoms detected on the B-BH16-1-treated semen coicis at 10 dpi or 20 dpi. The production of volatiles by the *P. palleroniana* strain B-BH16-1 significantly inhibited the growth of *A. flavus* on the semen coicis. The disease incidence among semen coicis was 0%, and the inhibition rate was up to 100%, in contrast to the control treatment.

Additionally, the relative abundance of aflatoxins in each treatment was also quantified. Five aflatoxins were detected in the control samples (AFB1, AFB2, AFG1, AFG2, and AFM1) by means of HPLC-FAD. At 10 dpi, the relative contents of AFB_1_, AFB_2_, AFG_1_, AFG_2_, AFM_1_, and AFs in the coix seeds were 61, 26, 268, 26, 1, and 355 μg/g, respectively. At 20 dpi, the relative contents of AFB_1_, AFB_2_, AFG_1_, AFG_2_, AFM_1_, and AFs in the coix seeds were 994, 4816, 3538, 9211, 2114, and 20,890 μg/g, respectively. While AFG_1_, AFG_2,_ and AFB_2_ were not detected in the coix seed samples treated with strain B-BH16-1 at 10 dpi or 20 dpi (Figure 4B–E). In addition, the expressions of three genes essential to aflatoxins synthesis were evaluated, and it was found that *P. palleroniana* strain B-BH16-1 significantly inhibited the expression of these genes (Figure 4F–H). *OMTB* and *aflS* were downregulated at 10 dpi and 20 dpi, and *aflR* was downregulated at 20 dpi. The expressions of *OMTB*, *aflS*, and *aflS* were decreased by 19, 9, and 4 fold, respectively, at 20 dpi. At the same time, the expressions of *OMTB* and *aflS* were reduced by 34 and 56 fold, respectively, at 10 dpi. Taken together, it can be concluded that *P. palleroniana* strain B-BH16-1 exerts vigorous biocontrol activity at the transcriptional level against aflatoxin biosynthesis in *A. flavus*.

### 2.5. Identification of Volatiles from Strain B-BH16-1

Volatiles from *P. palleroniana* strain B-BH16-1 were collected using a solid-phase micro-extraction (SPME) syringe and analyzed using the GC-MS/MS system. Three putative volatiles were detected in the volatiles of B-BH16-1 (Figure 5). These compounds, which had relative abundances (peak area to the total peak area of all compounds) greater than 3%, were identified as dimethyl disulfide (94 Da), 1-undecene (154 Da), and squalene (410 Da) (Figure 5B–D) through alignment with the National Institute of Standards and Technology (NIST) 17 MS database and the Willey MS databases. The retention times of dimethyl disulfide, 1-undecene, and squalene in GC-MS/MS were 11.390, 21.876, and 35.964 min, respectively. The relative abundances of dimethyl disulfide, 1-undecene, and squalene in GC-MS/MS were 3.99%, 59.71%, and 14.78%, respectively. The three compounds (dimethyl disulfide, 1-undecene, and squalene) were purchased as standards and tested for their antagonistic activity against mycelia and conidia of *A. flavus* to confirm their potential bioactivity.

### 2.6. Minimal Inhibitory Concentration Analysis of Volatiles from P. palleroniana Strain B-BH16-1 against A. flavus

The dimethyl disulfide, 1-undecene, and squalene standards were diluted to different concentrations, and FTF dual culture was performed with *A. flavus* in sealed petri dishes in order to test their antifungal activity. The results showed that dimethyl disulfide strongly inhibited *A. flavus* mycelial growth and conidia formation at concentrations ranging from 20 to 70 μL/L (compound volume/airspace volume). The inhibition rates ranged from 35 to 100%. The minimal inhibitory concentration (MIC) of dimethyl disulfide against mycelial growth and sporulation of *A. flavus* was 10 μL/L (Figure 6). Meanwhile, 1-undecene and squalene slightly inhibited the growth of mycelium and significantly inhibited spore formation at concentrations ranging from 20 to 70 μL/L. These results show that dimethyl disulfide, emitted from *P. palleroniana* strain B-BH16-1, is the main volatile inhibiting mycelial growth and conidia formation of *A. flavus*, while 1-undecene and squalene, emitted from *P. palleroniana* strain B-BH16-1, are the dominant volatiles inhibiting the conidia formation of *A. flavus*.

## 3. Discussion

*A. flavus* can not only cause corruption in crops and materials used in Chinese herbal medicine, it can also produce carcinogenic aflatoxins. Therefore, controlling these microbes is crucial critical in global production and storage systems for TCM and food. Here, we characterized the strain B-BH16-1 of the endophytic bacterial species *P*. *palleroniana* for use as a biocontrol agent to control the growth of aflatoxigenic *Aspergillus* pathogens. Biological assays and chemical profiling of the bioactive compounds revealed that three volatiles emitted by the *P. palleroniana* strain B-BH16-1 were responsible for potent inhibition of *Aspergillus* growth, spore formation, and aflatoxin production. This inhibitory activity was effective in both in vitro and in plant assays. Our results demonstrate that the volatiles produced by this microorganism are able to control the growth and spread of *A. flavus*, thereby reducing the associated aflatoxin contamination of TCM and crops and stored food/feed.

Dimethyl disulfide was one of the three most active components emitted by strain B-BH16-1, and completely inhibited the formation of spores and the growth of mycelia of *A. flavus*. Recently, researchers found this this component has previously been identified as possessing fungicidal, nematicidal, and herbicidal properties against plants [28,29] and other microorganisms such as *Bacillus cereus* [30], *Serratia* spp., *Bacillus* spp., and *Stenotrophomonas* spp. [31]. In addition, it can inhibit broad-spectrum to soil-borne phytopathogens such as *Meloidogyne* spp., *Fusarium* spp., *Phytophthora* spp., *Sclerotinia minor*, and *Rhizoctonia solani* [31,32]. Many studies have shown that dimethyl disulfide can be used as a fumigant to control soil-borne diseases [32,33]. These results provide a new idea for managing the mildew of TCM and food/feed in storage using fumigant measures.

Although the antifungal activities of 1-undecene and squalene were lower than that of dimethyl disulfide, their relative contents were higher than that of the dimethyl disulfide emitted from the *P. palleroniana* strain BBH16-1. Previous research has found that 1-undecene and squalene have antifungal and antinematode activities able to control soil-borne plant diseases. For instance, 1-undecene produced by *P. chlororaphis* and *P. fluorescens* have inhibitory activities on *Colletotrichum dematium*, *C. gloeosporioides*, *Fusarium oxysporum*, *F. subglutinans* and *Stemphylium lycopersici* in many plants, such as potato [34], maize [35], citrus [36], and olive [37]. In addition, squalene and 1-undecene can be used as a preservative for TCM and food/feed, because they have the ability to inhibit mold growth [38,39]. It is conceivable that a set of these compounds emitted from a single organism may generate synergistic antifungal activity. *P. palleroniana* strain B-BH16-1, which is capable of simultaneously synthesizing at least three volatile compounds, displays outstanding inhibitory activity against *A. flavus*. Further inhibition kinetics studies of different combinations of three volatile compounds against aflatoxigenic *A. flavus* and other agronomically important pathogens will provide new information on the efficient use of this strain and the mechanisms underlying inhibition.

It is well known that *A. flavus* mainly spreads through fungal spores in the air [40,41]. Compared with controlling *A. flavus* pollution by inhibiting mycelial growth and spore germination, blocking the spore’s formation of *A. flavus* is an effective strategy for preventing the contamination of *A. flavus* and its aflatoxins in food and materials used for TCM at the source. For example, the efficient prevention of the spread of *A. flavus* spores during the storage process of peanuts was achieved using plasmonic Ag-AgCl/α-Fe_2_O_3_ under visible light irradiation [40]. Our work found that dimethyl disulfide, 1-undecene, and squalene significantly inhibited the sporulation of *A. flavus* during the storage process of semen coicis, indicating that they can be used as an efficient potential strategy for preventing contamination with sporogenic fungi.

Aflatoxins are carcinogenic, teratogenic, and mutagenic secondary metabolites produced by *A. flavus.* In addition*,* when contamination with *A. flavus* is not visible, aflatoxins can be detected in food and materials used in TCM, even at levels exceeding the maximum permissible limit of aflatoxins. Our previous research found that aflatoxins in wheat increased with the extension of storage time, while the relative abundance of *A. flavus* did not change [42]. Therefore, we found that *P. palleroniana* strain B-BH16-1, which is able to simultaneously inhibit the growth of *A. flavus* and the production of aflatoxins via upregulation of genes essential to their synthesis, is of great significance for ensuring the safety of ingredients used in TCM and food. LC-ESI-MS analysis of aflatoxin content showed the inhibitory activity of *P. palleroniana* strain B-BH16-1 against aflatoxin production in semen coicis samples. The AFB1 and total AF contents of the untreated control semen coicis samples were higher than the maximum limits set by the Chinese Pharmacopoeia (20 and 5 ug/kg for total AFs and AFB1, respectively), the WHO (5 ug/kg for AFB1) and the European Union (4 and 2 ug/kg for total AFs and AFB1, respectively) [10]. When treated with B-BH16-1, aflatoxin production was inhibited entirely, and aflatoxins were undetectable using any of the methods used.

## 4. Conclusions

The *P. palleroniana* strain B-BH16-1 produces not only volatiles, but also diffusible antifungal substances. Many studies have shown that the *Pseudomonas* genera are able to produce a variety of diffusible substances that possess antifungal activity against multiple phytopathogens [43,44], such as lipopeptides [45,46]. The diffusible substances produced by the strain B-BH16-1 that exert inhibitory activity against *A. flavus* remain to be further analyzed. In conclusion, we identified an endophytic bacterium, *P. palleroniana* strain B-BH16-1, that produces three volatiles with inhibitory activity against mycelial growth and sporulation of *A. flavus*. The three bioactive volatiles produced by *P. palleroniana* strain B-BH16-1 are the well-known antimicrobial compounds dimethyl disulfide, 1-undecene, and squalene. Assays with semen coicis demonstrated that the volatiles emitted by *P. palleroniana* strain B-BH16-1 completely inhibited the growth of *A. flavus* and prevented aflatoxin biosynthesis. Microscopic observation revealed a severely damaged mycelial membrane and no formation of conidiophores. This newly identified endophytic plant bacterium strongly inhibits the sporulation and growth of *Aspergillus* and the synthesis of associated mycotoxins, not only providing valuable information regarding an efficient potential strategy for preventing *A. flavus* contamination in ingredients for TCM and food, but also acting as a potential reference in the control of toxic fungi.

## 5. Materials and Methods

### 5.1. Microorganisms and Plants

The strain B-BH16-1 used in this study was isolated from the tuberous root of *Pseudostellaria heterophylla* in Huangping County, Guizhou Province in China (N 107.9°, E 26.9°). Briefly, the tuberous root of *P. heterophylla* was collected for bacterial screening. The bacteria were isolated from the tuberous root using a dilution plate method on Luria–Bertani (LB) agar medium (5 g NaCl, 10 g tryptone, 5 g yeast extract per liter). The tuberous root was washed with ddH_2_O, sterilized in 75% ethanol for 30 s and 1% sodium hypochlorite for 6 min, and then washed with ddH_2_O for 30 s. Sterilized tuberous root samples were sonicated in 20 mL phosphate-buffered saline (PBS) solution for 30 min, and an aliquot (2 mL) of the suspension was diluted ten times. An aliquot (100 μL) of the dilution suspension was inoculated onto LB medium and cultured at 25 °C for two days, and then, individual colonies were isolated and stored at −80 °C in 20% glycerol. The isolated strains were collected and tested the antagonistic activity against *A. flavus* in dual culture and face-to-face (FTF) dual culture methods [16]. Bacterial strain B-BH16-1, with antifungal activity against *A. flavus*, was selected for further experiments.

Uniform-sized semen coicis (cv. Zheyi 1#) was purchased from local grocery stores. Semen coicis was placed in dishes and autoclaved at 121 °C for 20 min. The semen coicis was stored at 4 °C and used within two days. Phytopathogenic fungal *A. flavus* was isolated from diseased plants and used to detect the antifungal activity of strain B-BH16-1.

### 5.2. Identification of Strain B-BH16-1

*P*. *palleroniana* strain B-BH16-1 was cultured in LB medium and incubated in a shaker at 180 rpm at 28 °C in darkness for 48 h. The bacterial body was collected for DNA extraction as described previously [47]. The V3V4 fragment of rRNA sequences was amplified with universal primers 338F (5′-ACTCCTACGGGAGGCAGCAG-3’) and 806R (5′-GGACTACHVGGGTWTCTAAT-3’) [48]. The PCR reactions were conducted using the following conditions: pre-denaturation for 3 min at 95 °C, 27 cycles of denaturation for 30 s at 95 °C, annealing for 30 s at 55 °C, elongation for 45 s at 72 °C, and a final extension at 72 °C for 10 min. The PCR reactions were performed in triplicate in a 20 μL reaction mixture containing 4 μL of 5× FastPfu Buffer, 2 μL of 2.5 mM dNTPs, 0.8 μL of each primer (5 μM), 0.4 μL of FastPfu Polymerase, and 10 ng of template DNA. The PCR products were purified using an AxyPrep DNA Gel Extraction Kit (Axygen Biosciences, Union City, CA, USA) and then sequenced on an ABI 3730XL DNA Analyzer (Applied Biosystems, Waltham, MA, USA). The V3V4 sequences were aligned with the NCBI NR database by Nucleotide BLAST (https://blast.ncbi.nlm.nih.gov/Blast.cgi, accessed on December 30th, 2021) to determine the approximate phylogenetic affiliation of the strains. Taxonomy was confirmed on the basis of identity value with a threshold greater than 97% and the first ranked of all listed matches. The sequences of the strain B-BH16-1 were then blasted against the NCBI database (https://blast.ncbi.nlm.nih.gov/Blast.cgi, accessed on December 30th, 2021). Furthermore, the genome of B-BH16-1 was sequenced on an Illumina MiSeq platform (Illumina, San Diego, CA, USA) according to standard protocols by Majorbio Bio-Pharm Technology Co., Ltd. (Shanghai, China). ANI analysis was used to identify the taxonomy of strain B-BH16-1.

The biochemical activity of strain B-BH16-1 was identified using BIOLOG MicroStation™ System (Biolog Inc, Hayward, CA, USA) according to the method described by Gong et al. [16]. Briefly, a fresh B-BH16-1 clone was inoculated into IF-A GEN III Inoculating Fluid to OD600 = 0.95. The suspension was injected into GEN III MicroPlate and cultured at 37 °C for 12 h in darkness. The biochemical reaction of B-BH16-1 in each cell of MicroPlate was then aligned with the bacterial database in the BIOLOG MicroStation™ System.

### 5.3. Antagonistic Activity of Volatiles from Strain B-BH16-1 against A. flavus Mycelia

The antifungal activity of volatiles from *P*. *palleroniana* strain B-BH16-1 against *A. flavus* was determined using two sealed petri dishes (60 mm in diameter) as described previously [16,26]. The fresh *A. flavus* conidia were inoculated on a PDA (potato dextrose agar) center and challenged with strain B-BH16-1 in sealed airspace to detect the antifungal efficiency. Briefly, 10 μL fresh *A. flavus* conidia with 5 × 10^5^ CFU/mL concentration was inoculated to the center of one PDA plate. The other plate contained LB medium with fresh B-BH16-1 cells spread on the surface (100 μL, OD_600_ = 1.0) or ddH_2_O. The two dishes were placed FTF with a PDA plate above. All plates were cultured at 28 °C in darkness. After five days of incubation, the mycelial semidiameter of AF in each treatment was measured. The inhibition rate was calculated as follows: Inhibition rate (%) = (the mycelia semidiameter of control − the mycelia semidiameter of B-BH16-1 treatment)/the semidiameter of control × 100.

### 5.4. Control of A. flavus and its Aflatoxins on Semen Coicis Using Strain B-BH16-1

The biocontrol effect of strain B-BH16-1 against *A. flavus* on harvested semen coicis was tested in a sealed petri dish (0.2 L airspace). Autoclaved semen coicis (50 g) was inoculated with *A. flavus* conidial suspension (0.1 mL, 5 × 10^5^ CFU/mL) [16]. The samples were divided into two parts. One part was challenged with strain B-BH16-1 (the bacteria smeared on the LB plate at the bottom of the desiccator), and the other part, cultured in another desiccator, was challenged with a plain LB plate as a control. The experiment was conducted in independent triplicates. The semen coicis from treatments was collected 10 and 20 days post inoculation (dpi) and divided into two parts: one was stored at −80 °C for RT-qPCR analysis of aflatoxin biosynthesis genes, cell membrane synthesis genes, and sporulation genes, while the other part was dried at 60 °C for two days. The dried samples were milled for extraction and quantification of aflatoxins.

### 5.5. RNA Extraction and RT-qPCR Analysis

The semen coicis with *A. flavus* in the two treatments mentioned above was collected at 10 and 20 dpi and used for RNA extraction. Total RNA was extracted with Trizol reagent (Invitrogen) according to the manufacturer’s protocol. After DNaseI treatment, 2 μg of RNA was added to a 20 μL reaction system to synthesize first-strand cDNA using the Reverse Transcription System (Promega) according to the manufacturer’s instructions. Using 1.0 μL of 1:20 diluted cDNA as a template, PCR was performed in a 20 μL reaction volume with Bio-Rod SYBRII Super-Mix buffer on a Bio-Rad iQ2 PCR system (Bio-Rad, Hercules, CA, USA) [49,50].

Three specific genes involved in aflatoxin biosynthesis (including *OMTB*, *aflS*, and *aflR*), cell membrane biosynthesis (including *Csd1*, *Csd2*, and *Sam1*), and sporulation (including *RodB*, *FlbC*, and *FlbD*) of *A. flavus* were selected for RT-qPCR analysis (Table 1). β-tubulin gene was used as the endogenous control (reference gene) due to its relatively stable expression level. The relative quantification of gene expression changes was calculated using the 2^−ΔCt^ method.

### 5.6. Quantitative Analysis of Aflatoxins in Semen Coicis

The dried semen coicis samples with *A. flavus* were ground to a powder and passed through a No. 4 sieve. Aflatoxins in milled semen coicis were extracted using an organic reagent. Two grams of milled sample was dissolved in 10 mL acetonitrile/water (80/20, *v*/*v*), vortexed for 30 min, 0.5 g NaCl and 1 g MgSO_4_ were added, and then it was vortexed for 2 min. After centrifugation, 4 mL of supernatant was transferred to a new tube and mixed well with 0.05 g Octadecylsilyl (ODS) and 0.3 g MgSO_4_ for 2 min. To achieve re-centrifugation, 2 mL of supernatant was transferred to a new tube, dried with the nitrogen-blowing instrument, and dissolved with 50% acetonitrile. The sample was detected through HPLC-FLD (Shimadzu, Shanghai, China). Authentic reference standard aflatoxins (AFB1, AFB2, AFG1, AFG2, and AFM1), purchased from Sigma (Sigma-Aldrich, St. Louis, MO, USA), were used as standards for quantitative analysis. The limits of detection (LOD) for AFB1, AFB2, AFG1, AFG2 and AFM1 were 0.012, 0.009, 0.009, 0.012, and 0.015 ng/g, respectively. In addition, the limits of quantification (LOQ) for AFB1, AFB2, AFG1, AFG2 and AFM1 were 0.025, 0.019, 0.019, 0.025, and 0.027 ng/g, respectively. The relative content of AFs was calculated using the following formula: relative content of AFs = the detected content/(Sample weight × the expression of the β-Tubulin gene of *A. flavus* in the sample).

### 5.7. Microscopic Structural Analysis of A. flavus Cells and Sporulation when Affected by P. palleroniana Strain B-BH16-1

The microscopic structure of *A. flavus* cells and their sporulation beavior were examined using an OLYMPUS IX73 microscope (Olympus Corporation, Tokyo, Japan). The fresh *A. flavus* conidia were inoculated on the PDA plate center and challenged with strain B-BH16-1 in sealed airspace to detect the inhibitory effect on the microscopic structure of *A. flavus* cells and their sporulation behavior. A small sample was coated onto the slide to observe the mycelial growth and sporulation of *A. flavus* using an OLYMPUS IX73 microscope. In addition, we washed the *A. flavus* PDA plate with 5 mL ddH_2_O and took 10 μL in order to count the spore yield.

### 5.8. Volatile Identification and Inhibitory Analysis

B-BH16-1 was spread on the surface of LB medium in a 100 mL flask and cultured in darkness at 28 °C for 24 h. The produced volatiles were collected with a solid-phase micro-extraction (SPME) fiber containing divinylbenzene/carboxen/polydimethylsiloxane and detected using a gas chromatography–tandem mass spectrometry (GC-MS/MS) system (5977-7000D, Agilent Technologies Inc., Palo Alto, CA, USA) according to the methods described by Gong [26]. The experiment was performed three times. The compounds appearing in the B-BH16-1 profiles that were omitted in the control were considered the final volatiles. Authentic reference standard compounds were purchased and used in subsequent tests to detect the antifungal activity of volatiles produced by B-BH16-1. Fresh conidia (10 μL, 5 × 10^5^ CFU/mL) was inoculated to the center of the PDA plate. A paper disk on another petri dish was combined with authentic standard to final concentrations of 10, 20, 30, 40, 50, 60, and 70 μL/L (compound volume to airspace volume), respectively. Two dishes were sealed FTF and incubated at 28 °C as described previously. Disks with water added were used as control. The inhibition rate was calculated according to the method described above.

### 5.9. Data Analysis

All experiments were carried out at least in triplicate, and the results are reported as means ± standard deviations. The significant differences between mean values were determined using Student’s *t*-tests (*p* < 0.05) following one-way analysis of variance (ANOVA). Statistical analysis was performed using Origin software (Version 2018, OriginLab Inc., Northampton, MA, USA).

## Figures and Tables

**Figure 1 toxins-15-00077-f001:**
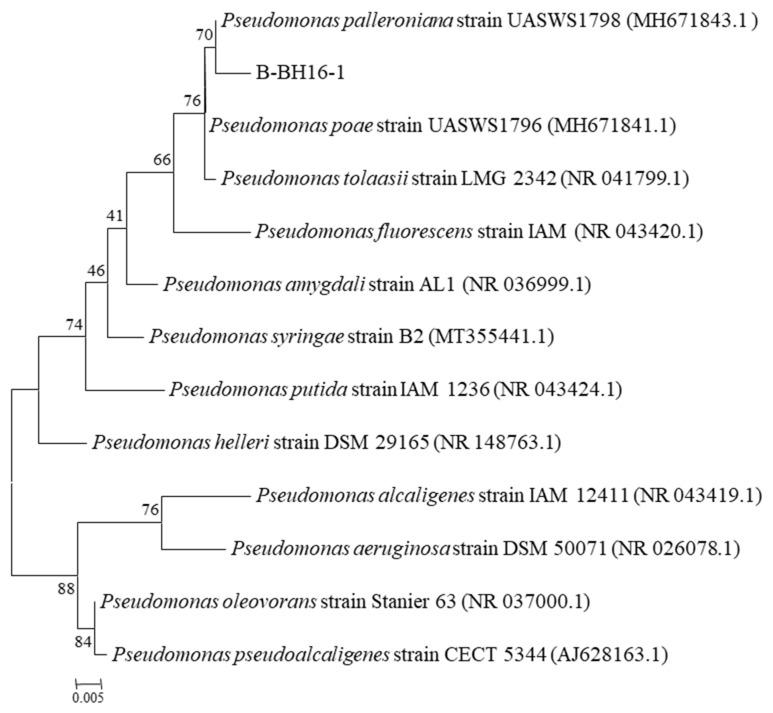
Phylogenetic tree of *P. palleroniana* strain B-BH16-1 and other homologous strains retrieved from GenBank database based on V3V4 fragment rRNA sequences. The tree was constructed using neighbor-joining methods, and the scale bar represents the number of substitutions per base position.

**Figure 2 toxins-15-00077-f002:**
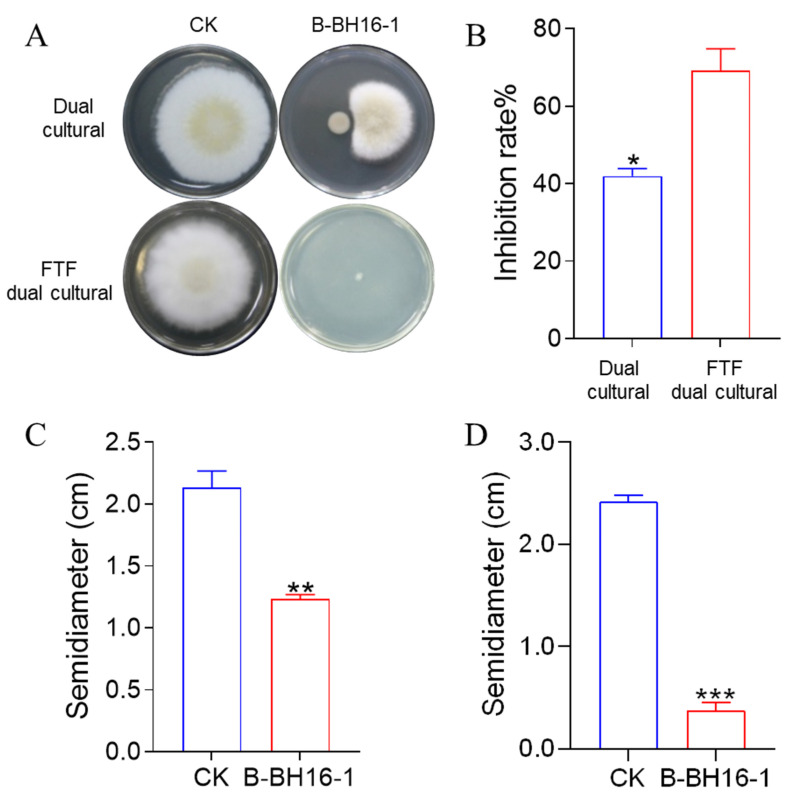
The inhibitory effect of volatiles from the *P. palleroniana* strain B-BH16-1 on *A. flavus*. (**A**) Growth morphology of *A. flavus* challenged by *P. palleroniana* strain B-BH16-1 in dual culture and FTF dual culture; (**B**) inhibition of *A. flavus* caused by the *P. palleroniana* strain B-BH16-1 in both tests; (**C**,**D**) growth semidiameter of *A. flavus* in dual culture and FTF dual culture, respectively. *, **, and *** refer to significant differences at *p* < 0.05, *p* < 0.01, *p* < 0.005.

**Figure 3 toxins-15-00077-f003:**
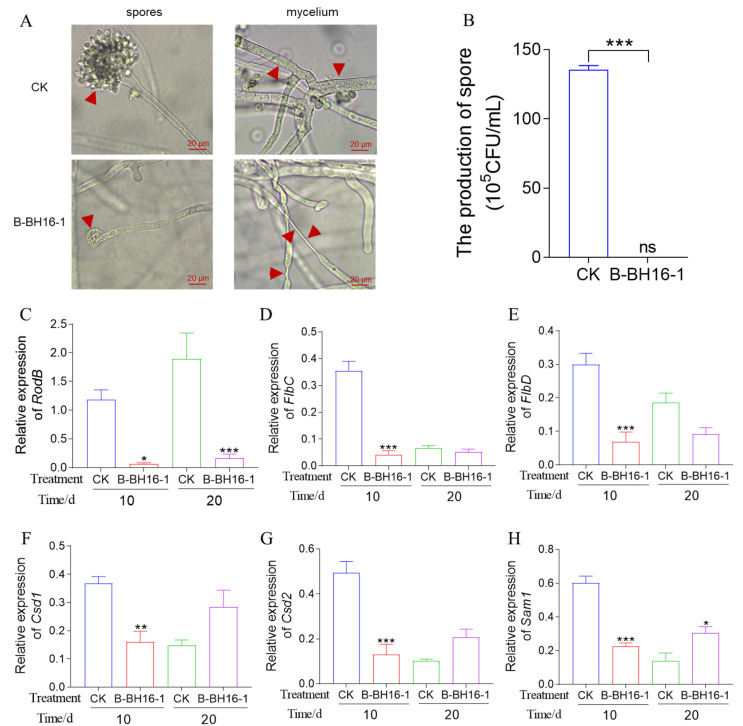
The inhibitory effect of volatiles from *P. palleroniana* strain B-BH16-1 against mycelium growth and sporulation of *A. flavus*. (**A**) Microscopic morphology of *A. flavus* challenged with B-BH16-1 in FTF dual culture; (**B**) conidia content of *A. flavus* challenged with B-BH16-1 in FTF dual culture; (**C**–**E**) relative expression of genes involved in the sporulation of *A. flavus* following treatment with volatiles from the *P. palleroniana* strain B-BH16-1; (**F**–**H**) relative expression of genes involved in cell membrane biosynthesis of *A. flavus* following treatment with volatiles from the *P. palleroniana* strain B-BH16-1. *, **, and *** refer to significant differences at *p* < 0.05, *p* < 0.01, *p* < 0.005.

**Figure 4 toxins-15-00077-f004:**
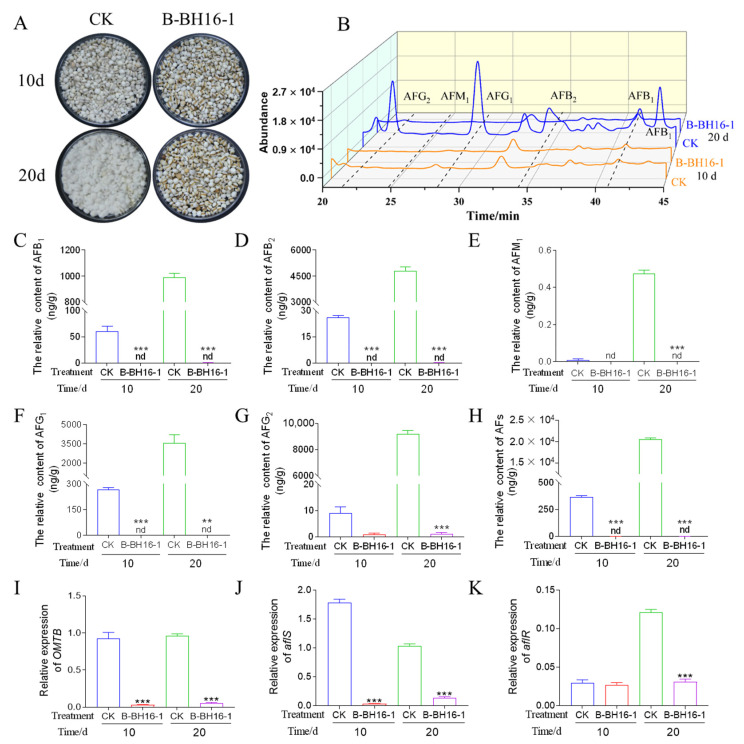
Biocontrol efficacy of volatiles from *P. palleroniana* strain B-BH16-1 against *A. flavus* and its aflatoxins on semen coicis. (**A**) Phenotypes of semen coicis infected with *A. flavus* in the presence (B-BH16-1) or absence (CK) of *P. palleroniana* strain B-BH16-1 at 28 °C for 10 and 20 days. (**B**–**H**) The contents of AFB1, AFB2, AFG1, AFG2, and AFM1 in semen coicis samples were detected using the HPLC-FAD system. (**I**–**K**) The relative expressions of genes involved in aflatoxin biosynthesis in *A. flavus* after treatment with volatiles from B-BH16-1. **, and *** refer to significant differences at *p* < 0.01, and *p* < 0.005. nd represent no detection.

**Figure 5 toxins-15-00077-f005:**
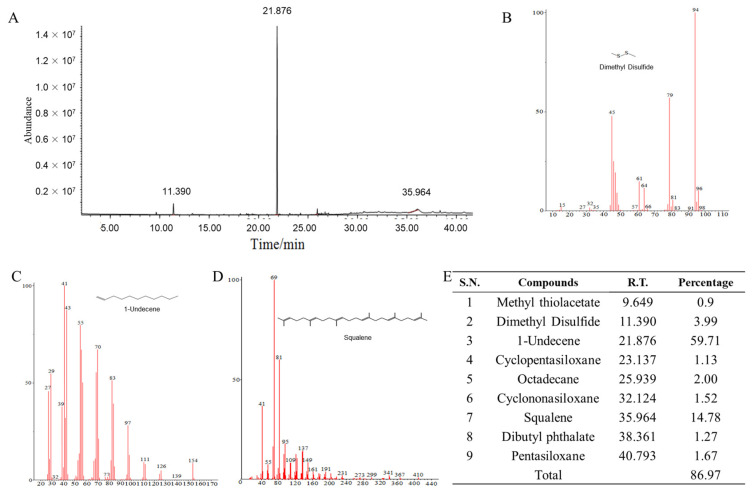
GC-MS/MS spectra of volatiles emitted from *P. palleroniana* strain B-BH16-1. Volatile compounds of B-BH16-1 cultured on an LB plate were detected at 48 hpi.

**Figure 6 toxins-15-00077-f006:**
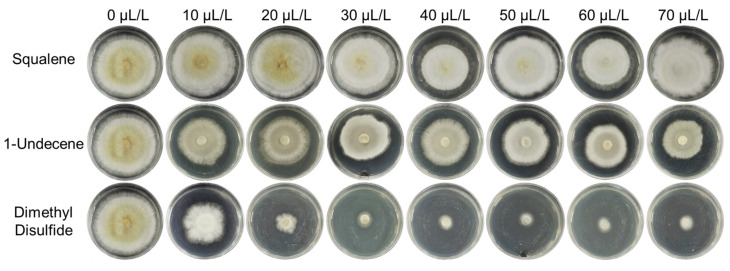
MIC analysis of the activity of volatiles against mycelial growth and sporulation of *A. flavus*. Three individual compounds, dimethyl disulfide, 1-undecene, and squalene, were assayed for their inhibitory activity against mycelial growth and sporulation on PDA for five days.

**Table 1 toxins-15-00077-t001:** Primers used in RT-qPCR analysis.

Gene Name	Gene Functions	Forward Primer	Reverse Primer
*NsdD*	sexual development transcription factor NsdD	CCGCCTATGAATACAGTGCTCCTAC	GACCCGCAAGTCCATTCCT
*FlbC*	C2H2 conidiation transcription factor FlbC	ACTGGTGAAAAGCGTAAGTTCTAAG	GAACCTTCTTGTGCCGACGA
*FlbD*	MYB family conidiophore development protein FlbD	ATGATTGAGCGGATGGTGAAC	GACCATTGAAAGTGCGTGAGAT
*Csd1*	C-3 sterol dehydrogenase--C-4 decarboxylase Csd1	ACTCCCTCTTCGCCCTTCA	AGCCACCGACCACCAAAAC
*Csd2*	C-3 sterol dehydrogenase--C-4 decarboxylase family protein Csd2	TTGGCAATGGGCAGAACC	CCGCTGGAAGTCCCAAAAG
*Sam1*	S-adenosyl-methionine-sterol-C- methyltransferase putative Sam1	ATGAAGAGTCCATTGCCGAACA	TCCTGGACGATGAAGCGGTT
*Csr1*	C-14 sterol reductase Csr1	CAACTCAACCCCGAACTTAGAC	GGATGTTCCACCCAAGCAAA
*Csr2*	C-14 sterol reductase Csr2	TCACTCGTGCCTTCCCTTTG	TAGACAGGGGAGCCCGACAA
*β-Tubulin*	Endogenous control, reference gene	TCTTCATGGTTGGCTTCGCT	CTTGGGTCGAACATCTGCT
*aflS*	*aflS*	GGTCGTGCATGTGCGAATC	GAGGGCAACAACCAGTGAGG
*aflR*	*aflR*	GCACCCTGTCTTCCCTAACA	ACGACCATGCTCAGCAAGTA
*OMTB*	*OMTB*	ATCAAGGAGACAGGTCCCCAA	GCAGTCCTTGTTAGAGGTGATA

## Data Availability

Data sharing not applicable.

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
