# Peer review of "Volatiles from Pseudomonas palleroniana Strain B-BH16-1 Suppress Aflatoxin Production and Growth of Aspergillus flavus on Coix lacryma-jobi during Storage"

_toxins, 2023, doi:10.3390/toxins15010077_

Round 1

Reviewer 1 Report

The article addresses an interesting topic, but you should consider some essential aspects:

1- Change the AF text to A. flavus, to avoid confusion with the aflatoxin abbreviation (lines 156, 163)

2- Line 155: enter the breakdown of PDA (potato dextrose agar)

3- What is the recovery percentage of aflatoxin extraction from real samples? How was the extraction technique validated?

4- How was the HPLC-FLD detection and quantification technique validated? Indicate in the text limit of quantification, the limit of detection...

5- Line 222: why were those concentrations of volatile compounds used?

6- Complete in figure 5 all the volatile compounds found with their respective retention rates (table)

Author Response

Dear reviewer,

We thank you very much for helping to improve the manuscript. We have studied all the comments and have incorporated changes to all those we can in this revised manuscript highlight blue. The followings are detailed point-to-point responses to the comments/suggestions made by you.

Comments and Suggestions for Authors

The article addresses an interesting topic, but you should consider some essential aspects:

1- Change the AF text to A. flavus, to avoid confusion with the aflatoxin abbreviation (lines 156, 163)

Response: We have changed the AF text to A. flavus in lines 169, 194, 221, 232, and 381.

2- Line 155: enter the breakdown of PDA (potato dextrose agar)

Response: We have made revision on that in line 369.

3- What is the recovery percentage of aflatoxin extraction from real samples? How was the extraction technique validated?

Response: That's a very good question. The recovery percentage of AFB1, AFB2, AFG1, AFG2 and AFM1 was 106.1%, 98.85%, 101.6%, 111.1% and 98.86% respectively, which were verified by QuEChERS according with Shen’s extraction method (Shen et al, 2022).

4- How was the HPLC-FLD detection and quantification technique validated? Indicate in the text limit of quantification, the limit of detection...

Response: That's a very good question. We analyzed the range, detection limit, quantification limit, precision, repeatability and stability of HPLC-FLD method. We found that the limits of detection for AFB1, AFB2, AFG1, AFG2 and AFM1 were 0.012, 0.009, 0.009, 0.012, and 0.015 ng/g, respectively. And the limits of quantification for AFB1, AFB2, AFG1, AFG2 and AFM1 were 0.025, 0.019, 0.019, 0.025, and 0.027 ng/g, respectively. We added the limit of quantification and the limit of detection in lines 417-420.

5- Line 222: why were those concentrations of volatile compounds used?

Response: We refer to Gong et al. 's study to set these concentrations (Gong et al, 2015).

6- Complete in figure 5 all the volatile compounds found with their respective retention rates (table)

Response: We have made revision on that in figure 5.

We are very sorry for our negligence. We appreciate for Reviewer’s warm work earnestly, and hope that the correction will meet with approval.

Once again, thank you very much for your comments and suggestions.

We are looking forward to your early response.

Reviewer 2 Report

Dear Editor,

In the MS the authors determined the antifungal and antimycotoxigenic effect of P. palleroniana strain against Aspergillus flavus. Besides, the authors evaluated the antifungal effect of volatiles compounds produced by P. palleroniana. In my opinion, the methods and results reported by authors related to antifungal and antimycotoxigenic activity of the strain are well conducted and described; therefore, I believe this MS could be accepted for publication after few modifications.

-I recommend authors to read text to avoid mistyping.

-The citation format is not correct.

-Authors can include a Conclusion section summarizing the results and the objectives accomplished.

Author Response

Dear reviewer,

We thank you very much for helping to improve the manuscript. We have studied all the comments and have incorporated changes to all those we can in this revised manuscript highlight green. The followings are detailed point-to-point responses to the comments/suggestions made by you.

Comments and Suggestions for Authors

In the MS the authors determined the antifungal and antimycotoxigenic effect of P. palleroniana strain against Aspergillus flavus. Besides, the authors evaluated the antifungal effect of volatiles compounds produced by P. palleroniana. In my opinion, the methods and results reported by authors related to antifungal and antimycotoxigenic activity of the strain are well conducted and described; therefore, I believe this MS could be accepted for publication after few modifications.

-I recommend authors to read text to avoid mistyping.

Response: We have made revision on that.

-The citation format is not correct.

Response: We have made revision on that.

-Authors can include a Conclusion section summarizing the results and the objectives accomplished.

Response: We have added the Conclusion section in line 300-317.

We are very sorry for our negligence. We appreciate for Reviewer’s warm work earnestly, and hope that the correction will meet with approval.

Once again, thank you very much for your comments and suggestions.

We are looking forward to your early response.

Reviewer 3 Report

The paper „Antifungal activity of volatiles emitted from Pseudomonas palleroniana strain B-BH16-1 against Aspergillus flavus and aflatoxins in Coix lacryma-jobi during storage“ is very well organized and clearly written. The methodology is adequate and the results are clearly presented with graphs and photographs and spectra.

 I have only two minor suggestions:

Change the title into „Volatiles from Pseudomonas palleroniana strain B-BH16-1 suppress aflatoxin production and growth of Aspergillus flavus on Coix lacryma-jobi during storage“

A small correction refers to line 440; instead of In summary, we identified an endophytic change… to In conclusion, …

Author Response

Dear reviewer,

We thank you very much for helping to improve the manuscript. We have studied all the comments and have incorporated changes to all those we can in this revised manuscript highlight orange. The followings are detailed point-to-point responses to the comments/suggestions made by you.

Comments and Suggestions for Authors

The paper “Antifungal activity of volatiles emitted from Pseudomonas palleroniana strain B-BH16-1 against Aspergillus flavus and aflatoxins in Coix lacryma-jobi during storage” is very well organized and clearly written. The methodology is adequate and the results are clearly presented with graphs and photographs and spectra.

I have only two minor suggestions:

Change the title into “Volatiles from Pseudomonas palleroniana strain B-BH16-1 suppress aflatoxin production and growth of Aspergillus flavus on Coix lacryma-jobi during storage”

Response: We have made revision on that in lines 1-3.

A small correction refers to line 440; instead of In summary, we identified an endophytic change… to In conclusion, …

Response: We have instead of “In summary, we identified an endophytic change…” to “In conclusion, …” in line 305.

We are very sorry for our negligence. We appreciate for Reviewer’s warm work earnestly, and hope that the correction will meet with approval.

Once again, thank you very much for your comments and suggestions.

We are looking forward to your early response.

Round 2

Reviewer 1 Report

It's a good job and important to the area. I suggest being accepted.